# Assessing the person-centered care framework and assessment tool (PCC-AT) in HIV treatment settings in Ghana: A pilot study protocol

Jessica E. Posner[1]*, Malia Duffy[2,3], Caitlin Madevu-Matson[1], Amy Casella[1], Henry Tagoe[4], Henry Nagai[4], Melissa Sharer[1,3]

1 International Division, JSI, Washington, DC, United States of America, 2 Health Across Humanity, LLC, Boston, MA, United States of America, 3 MPH Program, St. Ambrose University, Davenport, IA, United States of America, 4 International Division, JSI, Accra, Ghana

☯ These authors contributed equally to this work.
* Jessica_Posner@jsi.com

## Abstract

### Introduction

Evidence suggests that person-centered care (PCC) has the potential to overcome inequities in access to HIV services, support quality care that is responsive to diverse needs while increasing efficiencies and resilience of the health system. Despite emerging evidence on the effectiveness of PCC, there is limited information available on how to assess it in diverse clinical settings. This work builds upon a systematic literature review published elsewhere by this study team to develop a PCC framework for HIV treatment service delivery.

### Objectives

The PCC framework informed the development of the PCC assessment tool (PCC-AT) to assess the degree to which PCC activities are operationalized in diverse HIV treatment settings. The study objectives are to assess: (1) content validity of the PCC framework; (2) PCC-AT score consistency and reliability between health facility staff and clients; and (3) PCC-AT feasibility in HIV treatment settings.

### Methods

The study team will pilot the PCC-AT among staff in five health facilities and conduct subsequent focus group discussions (FGDs) to determine PCC-AT feasibility. Key informant interviews (KIIs) with clients will explore content validity among PLHIV relative to each subdomain of the PCC-AT and provide a basis to compare score concordance. Quantitative data among health facility staff will examine how many and which cadres participated in the PCC-AT pilot and FGD, years of experience, gender, and the time required to complete the PCC-AT. Information on clients will include total time accessing treatment at the study health facility, years since diagnosis, age and gender. Qualitative data analysis, using

**Data Availability Statement:** No datasets were generated or analysed during the current study. All

relevant data from this study will be made available upon study completion.

**Funding:** The author(s) received no specific funding for this work.

**Competing interests:** The authors have declared that no competing interests exist.

descriptive coding with NVivo or a similar software, will be drawn from transcripts from the PCC-AT pilots, FGDs and KIIs.

## Discussion

PCC assessment is a novel approach that aims to help health facilities assess and strengthen their ability to deliver PCC services to improve client outcomes.

## Introduction

HIV infections have reduced by 54% since the peak in 1996 and AIDS related deaths have reduced by 68% since 2004. [1,2] Among the estimated 38.4 million people living with HIV (PLHIV) globally, 25.6 million are in sub–Saharan Africa, followed by 6 million in Asia and the Pacific and 2.2 million in Latin America. [2] Despite progress over the past two decades, nearly 5,000 women and girls still become infected with HIV every week. [2] In addition, key populations (KP) including sex workers and their partners, men who have sex with men, people who inject drugs, and transgender people account for 70% of new infections globally. [2] Increased HIV risk for these populations is likely heightened by unique combinations of social, structural, and biological factors experienced at the individual level [3].

To achieve the UNAIDS goal to end AIDS by 2030, intermediate 2025 goals prioritizing person-centered strategies have been established as a means to expedite progress including comprehensive HIV services; people-centered and context-specific HIV services; and removal of societal and legal impediments to an enabling environment for HIV services. [4] The President's Emergency Plan for AIDS Relief (PEPFAR) defines person-centered care (PCC) as meeting the needs of individuals by increasing convenience, making services supportive and accessible, providing friendly services to diverse populations, and engaging communities and stakeholders. [5] The World Health Organization describes their vision for a person-centered framework as one in which "all people have equal access to quality health services that are co-produced in a way that meets their life course needs and respects their preferences, are coordinated across the continuum of care and are comprehensive, safe, effective, timely, efficient, and acceptable and all carers are motivated, skilled and operate in a supportive environment" [6].

Evidence suggests that PCC strategies have the potential to overcome inequities in access to HIV services, support quality care that is responsive to diverse needs while increasing efficiencies and resilience of the health system. [7] Practical examples of PCC within the context of HIV treatment available in the literature, including differentiated treatment delivery models, adolescent-friendly services, workplace interventions and peer-led support, have demonstrated that PCC can improve treatment outcomes across the care continuum [8–14].

### Development of the PCC framework for HIV treatment service delivery

Despite emerging evidence on PCC, there is scarce information available on how to define and operationalize it and how to assess it in diverse clinical settings. This study builds upon a previous systematic literature review published elsewhere by this team that established a framework for PCC for HIV treatment service delivery. [15] The initial framework identified three key domains (Staffing, Service provision, Direct client support) and 11 subdomains (respectively: composition, availability, competency; client feedback mechanisms, service efficiency and integration, convenience and access, digital health worker support tools; psychosocial support, logistical support, client agency, digital client support tools).

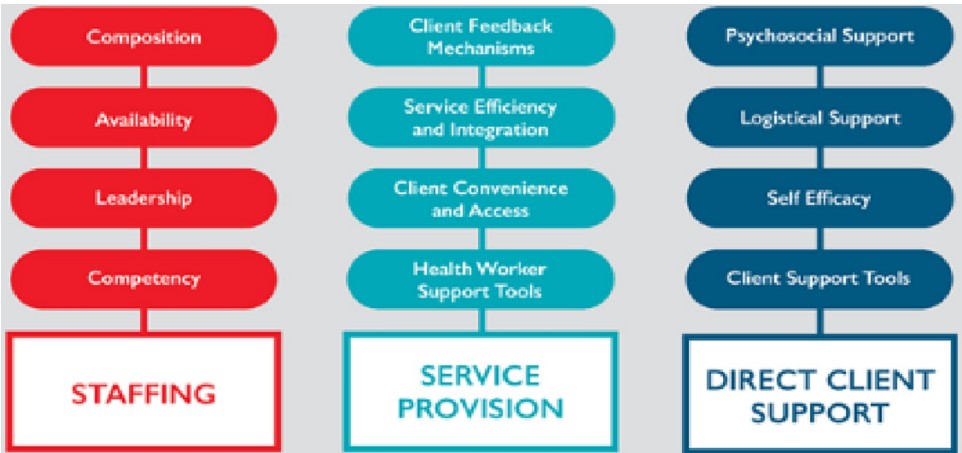

**Fig 1. Person-centered HIV treatment service delivery framework.**

Following publication of the systematic literature review, a process of content validity was employed to review the construct and appropriateness of the domains and subdomains and to identify any possible omissions. The PCC framework was presented at ICASA in December 2021 and the International AIDS Society conference (IAS) in Montreal, Canada in July 2022 where a focus group discussion was conducted to gather feedback from 25 HIV public health practitioners from seven countries (Ghana, Nigeria, Uganda, Zambia, Zimbabwe, Canada, and the United States) who were potential tool users currently implementing HIV service delivery programs in Zambia (USAID DISCOVER H and USAID Supporting and AIDSFree Era (SAFE) and Ghana (USAID Strengthening the Care Continuum). Additional feedback on the PCC framework was solicited from 50 IAS attendees via a tablet-based Google survey at the booth where the framework was posted.

Analyzing the feedback collected at the 2022 IAS conference, the study team identified framework gaps and strengths from stakeholder inputs which led to the formation of the twelfth subdomain of leadership under the Staffing domain–absent from the original framework, reinforcing the findings that PCC does not take place in a vacuum and requires leadership to be operationalized (Fig 1).

## Developing the person-centered care assessment tool (PCC-AT)

The PCC framework informed development of a person-centered care assessment tool (PCC-AT) to assess the degree to which clinics deliver person-centered care. To develop the assessment measure, the study team conducted three consultative rounds of review with three teams of five global experts from 4 countries (Zambia, Ghana, Uganda, the United States) in HIV, capacity strengthening, and strategic information. Following feedback from the team of global experts, the study team refined the PCC-AT and sought additional review from three monitoring and evaluation (M&E)/strategic information (SI) leads from health and HIV service delivery projects in Ghana and Zambia as well as in Uganda to gather insights into the practicality and applicability of the tool from an implementation perspective. M&E/SI insights were shared with the study team which helped to articulate performance expectations for each subdomain.

The process resulted in development of the PCC-AT, an Excel-based tool measuring health facility staff perspectives on PCC service delivery within HIV treatment settings. Like other tools widely used to measure organizational and technical performance of implementing

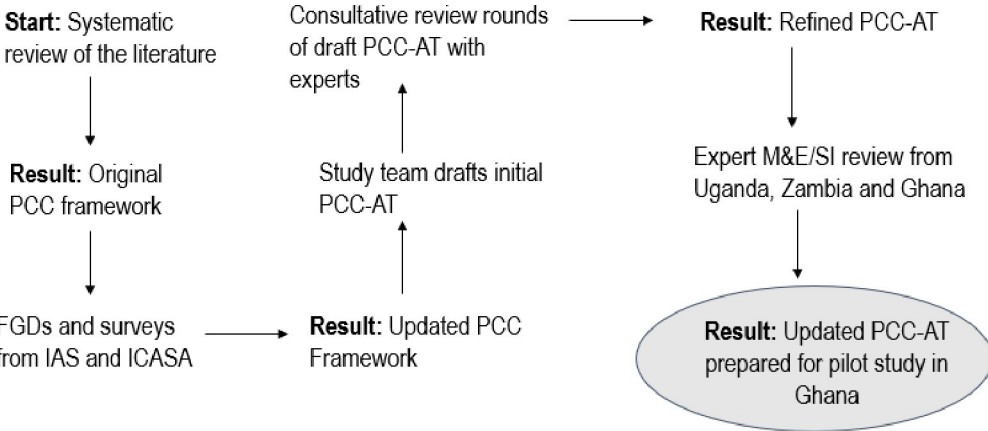

**Fig 2. Development of the PCC framework and the PCC-AT.**

partners, the PCC-AT includes drop-down menus and formulas to calculate composite scores from multiple tabs. There are 55 discrete performance expectations across the 12 subdomains. Fig 2 provides a description of how the PCC framework and the PCC-AT were conceived, developed and refined.

## Statement of objectives

**Study objectives** include assessing:

1. Assessing content validity of the PCC framework: Assessing the validity of the content for each sub-domain based upon the perspectives and experiences of PLHIV through key informant interviews (KIIs) will determine if the PCC framework content is valid. Gathering the perspectives of PLHIV is a critical measure given previous content validity assessments of the PCC framework were limited to public health practitioners.

2. Assessing PCC-AT scoring consistency and reliability: Assessing if the PCC-AT scores, identified by health facility staff, match the perspectives and experiences of PLHIV key informants who access services at the health facility.

3. Determining feasibility of the PCC-AT: Implementing the PCC-AT tool among health facility staff will be followed by FGDs to assess if the PCC-AT is useful, acceptable and implementable; how it is received by staff, any time or cost considerations, and if there are any required changes to enhance its implementation.

## Materials and methods

This is a protocol for a mixed methods study. Upon finalization, study results will be reported in-line with the Consolidated Criteria for Reporting Qualitative Research (COREQ) [16].

### Study setting

The USAID Strengthening the Continuum of Care project in Ghana was selected for this study due to: 1) availability of clinical sites supported by the project that are providing direct HIV treatment services; 2) expressed interest of project staff to scale-up PCC in an effort to strengthen services; and 3) availability of project staff already knowledgeable about the PCC-AT given their participation during the PCC framework and PCC-AT development

stages. Five health facilities were purposively sampled to represent geographical and size diversity in the levels of health facilities providing HIV treatment service in Ghana. Facilities include Regional Hospital (Effia Nkwanta Regional Hospital), two Government Hospitals (Kwesimintsim Government Hospital and Essikado Government Hospital), and two Quasi-Government health facilities (Shama Health Center and 2nd Medical Reception Station). This mix of facilities covers urban, parastatal, and community settings. The five participating facilities are located across three districts in the Western Region of Ghana spanning from the Ivory Coast (Comoé District) in the west to the Central region in the east. The region includes the capital and large twin city of Sekondi-Takoradi on the coast, coastal Axim, a large coastline, and a hilly inland area including Elubo, and covers an area of 13,842 sq. kilometers with a population of 2,060,585.[17] S1 File provides a listing of study health facilities and pre-identified alternate facilities that may be used, if required.

## Participants

**Health facility staff.**   We will aim for a minimum of seven health facility staff members to participate in the PCC-AT pilot and FGDs at each health facility for a total of five pilots and FGDs. The number of FGDs aligns with recent research estimating sample sizes in focus group research to achieve saturation. [18] The number of staff participants selected to participate in each PCC-AT pilot and FGD is based upon the study team's knowledge of staffing at study facilities and is consistent with the number of staff who will be available to participate on the study date while ensuring a mix of staff who can provide diverse perspectives based upon their roles.

Staff member selection will be purposive based upon their knowledge and availability to participate. Inclusion criteria for the PCC-AT pilot and FGDs include any staff who are head of department, ART providers, nurses, lab technicians, counselors, pharmacists (or the equivalent), data managers, and community group liaisons who are available on the day of data collection. Where there are more than seven eligible participants, we will encourage gender balance and diverse years of experience. Where possible the same participants who participated in the PCC-AT pilot will participate in FGDs immediately following the pilot.

**Client key informants.**   Recruitment of PLHIV for key informant interviews (KIIs) will be done using a convenience approach from a list of clients attending ART/HIV care services on the day of data collection. Following their appointment, clients will be invited to participate if they meet the following criteria: greater than 18, diagnosed with HIV, received ART/HIV services that day, and confirm that they have attended at least one appointment at the facility prior to the day of the visit. Any client under age 18 or attending their first ART visit will be excluded from the study. We will note commonalities between client responses to estimate how large a sample of clients are needed per facility. Drawing upon previous research, [19] we will determine that we have reached saturation when there is no new data after two consecutive interviews. At minimum, we will aim to include four clients per facility so as not to disrupt client flow.

## Study procedure

This study team includes two trained female Ghanaian researchers who are independent consultants who are trained in ethical research and are familiar with PCC concepts as well as the PCC-AT. They will be supervised by an implementation science technical advisor of the USAID Strengthening Care Continuum Project and will assess: (1) the content validity of the PCC framework among PLHIV; (2) scoring concordance reliability by comparing the PCC-AT scores assigned by health facility staff to the perspectives of PLHIV key informants;

and 3) feasibility of the PCC-AT through piloting the PCC-AT among health facility staff, followed by FGDs to assess staff perspectives on its implementation. We chose to conduct FGDs among health facility staff given their proven value in healthcare research to facilitate group exploration of systems processes and services while promoting synergies among participants and opportunities to disagree. [20] We chose to conduct KIIs among PLHIV key informants as a complementary measure to the FGDs. The individual structure of KIIs intends to protect clients' confidentiality and increase their confidence to share information on their experiences and to generate suggestions and recommendations in a private setting [21].

**PCC framework content validity.**   The study team will conduct KIIs among at least four PLHIV receiving services at each selected health facility, for an anticipated total of 20 PLHIV. Using a KII guide, the study team will gather insights on key informants' perspectives of their service experiences in-line with each subdomain within the PCC framework). Key informants will also be asked to express their degree of agreement regarding the importance of each of the PCC subdomains as it relates to HIV treatment service delivery (1–4; 1 = strongly disagree, 4 = strongly agree). They will also be asked to describe how activities within the subdomain should be adapted to improve their experiences of PCC. This exercise is a critical opportunity to gather insights from clients to ensure the content included in the PCC framework (which the PCC-AT is based upon) accurately encapsulates PCC delivery, capturing the experiences and knowledge from expert clients with lived experience (PLHIV).

**Score consistency and reliability.**   The study team will assess score consistency and reliability through a descriptive comparison of performance scores identified by health facility staff through the PCC-AT process against client-assigned performance scores for each PCC-AT domain, including average client scores by facility. This is important to better assess the concordance reliability between the providers scores and the actual lived experiences of PLHIV. A Spearman's rank correlation will be calculated to assess the statistical relationship between health facility staff and client scores.

**Feasibility.**   PCC-AT pilot implementation, led by a trained facilitator, will require health facility staff to discuss each performance expectation and use a specified benchmarking approach (e.g., simple yes/no or mostly/partly/none) coded for each performance expectation to respond. The study team anticipates that completion of the tool will require 90-minutes. Once the scoring is complete, each PCC-AT pilot will culminate in an action planning process, led by the facilitator, based upon PCC-AT scores. Through the action planning process, as the final step of the PCC-AT conducted among the same health facility staff, participants identify areas of low performance and strengthening activities that address performance gaps.

Following implementation of the PCC-AT, staff perspectives on its feasibility and utility will be elicited through one FGD at each health facility. The study team will use a FGD guide to collect qualitative data on users' experiences of the process (e.g., length of time, ease of use, comprehensibility), reactions to results and content (e.g., relevance, comprehensiveness, ability of the tool to assess PCC), and perceptions of the tool's usability in the context of the health facility workplace. PCC-AT implementation and FGDs will take place in unused health facility spaces including empty waiting rooms, meeting rooms or treatment rooms.

## Informed consent

For study participants who are literate, the consent form, in English, will be given to them to read and ask question(s) as needed prior to providing their signature. For participants who cannot read or write, consent will be sought in the presence of a chosen witness by the participant. The consent forms will be translated by the interviewee into the language that the participant can best understood and express themselves in. Twi and Fanti are the dominant

languages of the study districts and English is the official language of the country. Language will not be used as an exclusion criterion. The client will sign the informed consent sheet in the presence of a self-selected witness. In addition, prior to each PCC-AT pilot and FGD, the study team will distribute a study summary sheet and obtain written consent from each health facility participant. Participants will sign consent forms in front of a witness of their selection.

## Materials

In addition to the PCC-AT tool, materials will include the FGD guide (S2 File) and the KII guide (S3 File). Translation of the PCC-AT and FGD tool will not be required due to broad fluency in English among health facility staff. To ensure that clients can provide feedback, the KII will be translated into the local language Twi. The KIIs will be conducted in either English or Twi, based upon each participant's preference. All FGDs and KIIs will be recorded and transcribed. KIIs in Twi will be translated into English after transcription. Informed consent for health facility staff (S4 File) will be available in English and for PLHIV in English and Twi (S5 File), based upon individual client preference.

## Data analysis plan

Data analysis will include all four sources including the: (1) KII recording and transcripts, (2) PCC-AT pilot discussion recording and transcripts; (3) FGD recordings and transcripts; and (4) PCC-AT composite score page. Descriptive statistics will be gathered from key informants on age, gender, time (years/months) accessing services at the health facility, time on treatment, and year diagnosed with HIV. Descriptive statistics will also be gathered from health facility staff participants including age, gender, position, and length of time in employment at the health facility.

Quantitative analysis will include information collected on how many and which staff (by cadre) participated in the PCC-AT pilot and FGD, their years of experience at the health facility, gender, and the time required to conduct the tool. For KII participants, we will include information collected on total time accessing treatment services at the study health facility, years since diagnosis, age and gender. The study team will assess score consistency and reliability through a descriptive comparison of performance scores identified by health facility staff through the PCC-AT process against client-assigned performance scores for each PCC-AT domain, including average client scores by facility. A Spearman's rank correlation will be calculated to assess the statistical relationship between health facility staff and client scores.

Qualitative analysis will derive from discussions associated with PCC-AT pilot and the resulting action plans, FGDs, and KIIs and consist of thematic analysis of findings. For each data source, one researcher will examine findings and develop an initial codebook using NVIVO software which will be shared with a second researcher. PCC-AT pilot and action planning discussions, and FGD and KII transcription data will be separately analyzed by the two qualitative researchers using the codebook as the foundation while adding new codes and sub-codes as themes emerge through further analysis. Illustrative axial codes the study team anticipates for the FGDs will include length of time, structure, comprehensibility, and suggestions for changes. For the KIIs, illustrative axial codes that the study team anticipates will include staff accessibility, wait times, client feedback, ART dispensation, and logistical support. After both qualitative researchers separately analyze the data, they will meet to compare data included under each code and sub-code. Differences will be identified and discussed. If consensus is not reached, a third study team member will help to make final determinations.

## Data security

Participant confidentiality will be protected throughout study procedures including, 1) not including any names of participants (health facility staff or PLHIV) on any data collection materials, 2) storing data in a secure place, 3) developing codes during qualitative data analysis so that any potential identifiers (e.g., staff positions) are not associated with data, and 4) maintaining strict adherence to the principles of voluntary participation, confidentiality, anonymity and protection of human subjects as guaranteed by the consent form.

## Study status and timeline

Data collection in Ghana will take place over the month of May 2023 and data analysis will take place in June and July 2023. Study findings will be disseminated near September 2023.

## Ethical approval and protocol registration

The study has received ethical approval from JSI's IRB (IRB #22-53E) and Ghana Health Service Ethics Review Committee (Protocol ID NO: GHS-ERC 008/10/22). The protocol was registered with OSF prior to data collection (https://doi.org/10.17605/OSF.IO/763M4).

# Discussion

## Limitations

Conducting research in HIV clinical sites is complex, as there are many factors that impact the delivery of care and the collection of data which may impact the success of facility-based interventions. Implementation science seeks to understand and influence how scientific evidence is put into practice for health improvement. Piloting the PCC-AT includes measuring scores on the three domains and 12 subdomains included in the tool which could take away time devoted to providing care which is a challenge with implementation research.

The limitations of the present study are linked to the small sample of facilities and staff who will participate, which can limit the power of the study. However, these smaller sample sizes are appropriate and will help the authors determine if larger scale studies are warranted. Additionally, content validity is one of the weakest forms of validity, therefore this study will be an important first step to establish the validity of the PCC framework which informed the domains and subdomains of the PCC-AT. However, this study addresses an important limitation from previous studies in the fact that it includes the knowledge and lived experiences from individuals who identify as a person living with HIV. PLHIV will be recruited to provide their perspectives on PCC framework content and scores from the PCC-AT. Still, the number of PLHIV is small and ungeneralizable therefore limiting external validity and the interpretation of these results. In addition, the researcher's presence during qualitative data collection may affect/bias participant's responses and clients who consent to participate in the study may offer different inputs than any clients who may decline to participate. As convenience sampling was used it can result in a biased and unrepresentative of the population in question. The perceptions of the clients were recorded on a four-point Likert Scale which may not sufficiently capture the perspectives and experiences of key informants. The quality of the data collected is dependent on the researcher's skills, background, and personal biases. In effort to improve the rigor all interviews were done by trained Ghanian researchers fluent in the three main languages in Ghana. We have also chosen to limit the study to people aged 18 and above due to parental consent challenges which limits the perspectives that will be gathered. Though client key informants will be reassured that their responses will be confidential and that there will be no positive or negative consequences to their participation, there remains a risk of

response bias if clients do not feel secure sharing information with the study team. Similarly, health facility staff may not openly share their perspectives in a focus group setting. Qualitative data captured via the key informant guide aims to overcome this challenge through further drawing out information via one-on-one interviews.

## Dissemination

Given the potential of the PCC-AT to assess PCC within diverse HIV treatment settings and the demonstrated potential of PCC to improve client outcomes [22,23] knowledge sharing will be prioritized by the study team following the study. This will include convening a meeting at IAS/ICASA 2023 on the topic to present findings and share implementation experiences including potentially hosting a training with global partners on its use. Findings will also be shared with the HIV Technical Working Group, Ghana AIDS Commission Research, Monitory and Evaluation Committee and the HIV community during the National HIV and AIDS Conference (NHARCON). We also anticipate publishing findings from the Ghana study in September 2023.

Information from the study will be used to further refine the PCC-AT including its content, indicators, scoring, and application in practice. Once finalized, the study team will collaborate with the 99 health facilities under the purview of JSI projects in Ghana to integrate the tool as a standard of care. The study team will also use findings from the Ghana study to inform a planned study in Zambia in August 2023 that will evaluate the relationship between a health facility's PCC-AT score and key health indicators including antiretroviral treatment (ART) coverage, continuity in ART, and viral suppression.

## Supporting information

**S1 File. Selected and alternate study health facilities.**
(DOCX)

**S2 File. FGD guide.**
(DOCX)

**S3 File. KII guide.**
(DOCX)

**S4 File. Informed consent for health facility staff.**
(DOCX)

**S5 File. Informed consent KII.**
(DOCX)

## Acknowledgments

The authors would like to express gratitude to their colleagues and other individuals who have provided thoughtful insights to strengthen the PCC HIV treatment framework as well as the PCC-AT including Antonia Powell (Co-director of JSI's HIV/Infectious Disease Center), Dr. Yaw Ofori Yeboah (Regional Director of Health Service, Western Region), and Yussif Ahmed Abdul Rahman, Dr. Michael Deledem Kwashie, Paul Yikpotey and David Tetter Nartey, Edward Adiibokah of the USAID Strengthening the Care Continuum Project.

## Author Contributions

**Conceptualization:** Jessica E. Posner, Malia Duffy, Caitlin Madevu-Matson, Henry Tagoe.

**Investigation:** Jessica E. Posner.

**Methodology:** Jessica E. Posner, Malia Duffy, Caitlin Madevu-Matson, Amy Casella, Henry Tagoe, Melissa Sharer.

**Project administration:** Jessica E. Posner, Henry Nagai.

**Writing – original draft:** Jessica E. Posner, Malia Duffy.

**Writing – review & editing:** Jessica E. Posner, Malia Duffy, Caitlin Madevu-Matson, Amy Casella, Henry Tagoe, Henry Nagai, Melissa Sharer.

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
