## [Decision Letter · Decision Letter 0]

17 Sep 2023

PONE-D-23-15863The person-centered care assessment tool (PCC-AT): A study protocol to assess content validity, score consistency and reliability, and feasibility in HIV treatment settings in GhanaPLOS ONE

Dear Dr. Posner,

Thank you for submitting your manuscript to PLOS ONE. After careful consideration, we feel that it has merit but does not fully meet PLOS ONE’s publication criteria as it currently stands. Therefore, we invite you to submit a revised version of the manuscript that addresses the points raised during the review process.

After reviewing the manuscript and the reviewer's comments, I do believe that addressing or clarifying each of the issues raised by reviewers would strengthen the paper. Both reviewers also suggested some strengthening and clarification of the qualitative methods and a reflection of qualitative limitations. I additionally recommend adding citations to the qualitative methods and reviewing the CORE-Q guidelines to add any details on qualitative data collection, analysis and/or rigor that are currently missing from the paper:  https://www.equator-network.org/reporting-guidelines/coreq/ 

We look forward to receiving your revised manuscript.

Kind regards,

Emily A Hurley, M.P.H., Ph.D.

Academic Editor

PLOS ONE

Journal Requirements:

Reviewers' comments:

Reviewer's Responses to Questions

**Comments to the Author**

1. Does the manuscript provide a valid rationale for the proposed study, with clearly identified and justified research questions?

Reviewer #1: Yes

Reviewer #2: Yes

2. Is the protocol technically sound and planned in a manner that will lead to a meaningful outcome and allow testing the stated hypotheses?

Reviewer #1: Yes

Reviewer #2: Partly

3. Is the methodology feasible and described in sufficient detail to allow the work to be replicable?

Reviewer #1: No

Reviewer #2: No

4. Have the authors described where all data underlying the findings will be made available when the study is complete?

Reviewer #1: No

Reviewer #2: Yes

5. Is the manuscript presented in an intelligible fashion and written in standard English?

Reviewer #1: Yes

Reviewer #2: Yes

6. Review Comments to the Author

You may also provide optional suggestions and comments to authors that they might find helpful in planning their study.

Reviewer #1: July 14, 2023

Patient-centered care is critical to improving the well-being of people living with HIV. This manuscript tackles a vital topic in HIV service delivery. Its strengths include:

• Attempt to create a scale to measure patient-centered care.

• Use of a systematic review-informed theoretical framework.

• Sampling across multiple facilities in Ghana.

There are a few areas that need strengthening.

Introduction

o The authors describe that they collected feedback on the domains of the scale at the IAS and ICASA conferences. It would be helpful to include the method of data collection. Figure 2 shows that they used FGDs and surveys from IAS and ICASA. However, these methods are not mentioned in the introduction.

o The study objectives need to be rewritten as clear sentences instead of short phrases. For example: Determine the feasibility of PCC-AT instead of “PCC-AT feasibility.”

o Relating to the first objective, is it a re-evaluation of the content validity of the PCC-AT, given the data they collected from the IAS and ICASA? It is hard to tell because details of what was collected in the FGDs and surveys at the IAS meeting ICASA are scarce.

o One argument advanced in this paper is that PCC may differ across contexts or clinical settings. So, what context is this scale designed for? Ghana? All African countries? Global tool? Why was content validity data from the first round collected from many countries, and now data is being collected from Ghana alone?

Materials and Methods

o The sampling of health workers and participants could be more robust and not tied to the day of data collection. It appears like, if the research team shows up, whoever is there is who they interview, particularly health workers. Because the scale includes multiple domains, they will need to be more purposeful in selecting participants. If specific groups of participants are away on the data collection day, should more attempts be made to recruit them?

o One of the domains is leadership, but in the list of participants to be interviewed, there appears to be no purposeful attempt to interview leaders, heads of departments, units, or staff.

o The justification for seven people needs to be clarified. Any reason why? Particularly as they also desire a balance in gender and years of experience.

o Is there a reason why adolescents are excluded?

o It also needs to be clarified why those working less than 12 months are excluded. Wouldn’t sampling them give a continuum of years of experience?

o It would be helpful for the authors to include cognitive interviewing to understand the wording of the items within the scale. Ghana has many languages, and research assistants sometimes cannot read and write in the local language. They orally translate English into the local language. So they must figure out the precise wording of items and assess how participants understand the questions.

o Having worked on scales in Ghana, there is a real challenge when a Likert scale is used as a response format. People sometimes have difficulty distinguishing between a range of categories. In addition, there are often responses where participants are non-committal to any specific response. They will say: “It is “OK”—meaning neither good nor bad. Thus, there needs to be a neutral response to the options provided to participants. It would be compelling and helpful if they also asked about the scoring format for the scale.

o Line 200: “Once the scoring is complete, each PCC-AT pilot will culminate in an action planning process, led by the facilitator, based upon PCC-AT scores.” Could they clarify who the participants here are? What will they be planning?

o Line 179—relating to the analysis “ feasibility of the PCC-AT through a pilot among health facility staff followed by FGDs.” I was unclear if the pilot is different from the FGDs. Are the health workers scoring first, and then the data is used for the FGDs, or the FGDs is part of the pilot testing, or are they done in the same session?

o The challenge is often finding a place at the health facility to interview privately. Is there a designated room for this?

o Could they be specific on the statistical analysis to assess the “concordance reliability by comparing the PCC-AT scores assigned by health facility staff to the perspectives of PLHIV key informants”? What statistics would be reported? How would concordance be evaluated?

o Perceived and enacted stigma is integral to the lived experience of people living with HIV. Is there a specific sub-domain that captures stigma?

o Would there be room in the methodology for iteration? Like participants would suggest various modifications. Would those modifications be made, and feedback solicited on them before the scale is finalized? It seems like all the data would be collected and analyzed later. If there is room to start with a few participants, refine, and sample again—like how Grounded Theory works—so that the final product is complete.

Reviewer #2: Summary:

This article presents the protocol for a study assessing content validity, score consistency and reliability, and feasibility in HIV treatment settings in Ghana of a person-centered care assessment tool (PCC-AT). It is an original and timely study protocol with growing attention to person-centered care approaches. However, there are concerns, particularly to the methods and materials section, and the goal of the protocol, which reads as a mix of framework/theory development, measure assessment, and implementation assessment.

Title: The title is a bit misleading. I presumed based on the title that this protocol would focus on some standard quantitative psychometric approaches to assessing reliability. In addition, an entire objective of the study is focused on content validity of the PCC framework, not the PCC-AT.

Introduction:

• Page 4, line 87 says six countries but lists seven

• Page 5, line 110 says give countries but lists four

Methods:

• I am a bit unclear about the assessment of the content validity of the PCC framework and timing. I would think that assessment of the framework for content validity would proceed finalization of the PCC-AT (and any assessments of reliability or validity). Either a clarification of timing or further explanation is needed.

• Overall, some reorganization of this section would help to make it clear. For example, you could organize the different activities under the “Participants” sub-section by type of data collection, or by objective. Inclusion/exclusion criteria for each activity would also be easier to read if bulleted.

• The methods for the focus groups are not well described. Is it one per facility? How will they be convened? Any comment/justification on sample size?

• More explanation/justification of choice of methods to address the objectives is needed.

• Is the PCC-AT being administered to KII participants in full. This is unclear as described.

• Are the authors able to share the PCC-AT? Methods overall are hard to follow without that as a grounding instrument to look at. For example the authors say as part of the KII that participants will score each domain: is that part of the PCC-AT? What is the criteria for scoring a domain as “Very poor” “Poor”…etc? I can see how the authors are addressing content validity potentially, but more explanation is needed to say there is an assessment of reliability. I do not see a description of data collection or anlaysis that assess reliability (e.g. inter-rater, test-retest…etc.)

• Page 8, line 171: authors say they “will note commonalities between client responses to estimate how large a sample of clients are needed per facility to reach saturation”. Authors should consider the following suggestions to strengthen sample size justification:

o Define/include appropriate citations for the principle of saturation.

o Better explain how they will assess saturation to ensure it has been achieved.

• Page 8, line 175: please add additional details about the trained researchers (are they Ghanian, American, members of the investigative team?)

• Please add additional details on informed consent

o What languages will the consent form be available in or will it be translated from English by the person obtaining consent?

o What about people who cannot write?

o What if people do not speak Twi or English (as noted under Materials) – will this be an exclusion reason?

• Please add more details (and relevant citations) to the qualitative analysis plan, at a minimum:

o More description of codebook development

o Software used or approach to applying codes

o Domains you intend to structure analysis under

o Will you do formal intercoder comparison? If so, describe methods and include citations. You not have to do intercoder comparisons, but if not, describe process for aligning coding and harmonizing interpretation of codes

o More description of coding and code comparison/how discrepancies will be handled

Discussion

• Is there discussion of other practical study considerations not covered in other sections the authors could include here?

• Limitations:

o This study is largely qualitative, but the limitations read as if written for a quantitative study (e.g. external validity). Authors should rather consider appropriate qualitative measures of rigor to comment on or add to this section.

• Dissemination

o Please describe any dissemination plans to PLHIV and/or study participants

o The authors say the PCC-AT tool will be disseminated and integrated into standard of care. Will there be no formal psychometrics, cognitive interviewing…etc involved beforehand? If so, this would be worth noting as a limitation of the assessment described in this protocol

7. PLOS authors have the option to publish the peer review history of their article (what does this mean?). If published, this will include your full peer review and any attached files.

Reviewer #1: No

Reviewer #2: No

---

## [Author Response · Author response to Decision Letter 0]

9 Oct 2023

We have provided a rebuttal letter and made edits to the helpful feedback

---

## [Decision Letter · Decision Letter 1]

30 Nov 2023

Assessing the person-centered care framework and assessment tool (PCC-AT) in HIV treatment settings in Ghana: A pilot study protocol

PONE-D-23-15863R1

Dear Dr. Posner,

We’re pleased to inform you that your manuscript has been judged scientifically suitable for publication and will be formally accepted for publication once it meets all outstanding technical requirements.

Kind regards,

Emily A Hurley, M.P.H., Ph.D.

Academic Editor

PLOS ONE

Additional Editor Comments (optional):

Reviewers' comments:

Reviewer's Responses to Questions

**Comments to the Author**

1. Does the manuscript provide a valid rationale for the proposed study, with clearly identified and justified research questions?

Reviewer #1: Yes

Reviewer #2: Yes

2. Is the protocol technically sound and planned in a manner that will lead to a meaningful outcome and allow testing the stated hypotheses?

Reviewer #1: Yes

Reviewer #2: Yes

3. Is the methodology feasible and described in sufficient detail to allow the work to be replicable?

Reviewer #1: Yes

Reviewer #2: Yes

4. Have the authors described where all data underlying the findings will be made available when the study is complete?

Reviewer #1: Yes

Reviewer #2: Yes

5. Is the manuscript presented in an intelligible fashion and written in standard English?

Reviewer #1: Yes

Reviewer #2: Yes

6. Review Comments to the Author

You may also provide optional suggestions and comments to authors that they might find helpful in planning their study.

Reviewer #1: The response were adequate. just one minor edit when they are editing their proofs.

In the introduction section, under study objectives, line 132, the word "assessing: should perhaps be deleted as it is repeated in the other objectives.

Reviewer #2: Thank you to the authors for their revisions. I look forward to seeing the results of this work in future publications.

7. PLOS authors have the option to publish the peer review history of their article (what does this mean?). If published, this will include your full peer review and any attached files.

Reviewer #1: No

Reviewer #2: No

---

## [Editor Report · Acceptance letter]

28 Dec 2023

PONE-D-23-15863R1 

PLOS ONE

Dear Dr. Posner, 

I'm pleased to inform you that your manuscript has been deemed suitable for publication in PLOS ONE. Congratulations! Your manuscript is now being handed over to our production team.

Kind regards, 

on behalf of

Dr. Emily A Hurley 

Academic Editor

PLOS ONE